# Curcumin and Diclofenac Therapeutic Efficacy Enhancement Applying Transdermal Hydrogel Polymer Films, Based on Carrageenan, Alginate and Poloxamer

**DOI:** 10.3390/polym14194091

**Published:** 2022-09-29

**Authors:** Katarina S. Postolović, Milan D. Antonijević, Biljana Ljujić, Slavko Radenković, Marina Miletić Kovačević, Zoltan Hiezl, Svetlana Pavlović, Ivana Radojević, Zorka Stanić

**Affiliations:** 1Department of Chemistry, Faculty of Science, University of Kragujevac, 34000 Kragujevac, Serbia; 2Faculty of Engineering and Science, School of Science, Medway Campus, University of Greenwich, Chatham Maritime, Kent ME4 4TB, UK; 3Department of Genetics, Faculty of Medical Sciences, University of Kragujevac, 34000 Kragujevac, Serbia; 4Department of Histology and Embryology, Faculty of Medical Sciences, University of Kragujevac, 34000 Kragujevac, Serbia; 5Department of Biology and Ecology, Faculty of Science, University of Kragujevac, 34000 Kragujevac, Serbia

**Keywords:** curcumin, diclofenac, films, biopolymers, carrageenan/alginate/poloxamer, wound healing

## Abstract

Films based on carrageenan, alginate and poloxamer 407 have been formulated with the main aim to apply prepared formulations in wound healing process. The formulated films were loaded with diclofenac, an anti-inflammatory drug, as well as diclofenac and curcumin, as multipurpose drug, in order to enhance encapsulation and achieve controlled release of these low-bioavailability compounds. The obtained data demonstrated improved drug bioavailability (encapsulation efficiency higher than 90%), with high, cumulative in vitro release percentages (90.10% for diclofenac, 89.85% for curcumin and 95.61% for diclofenac in mixture-incorporated films). The results obtained using theoretical models suggested that curcumin establishes stronger, primarily dispersion interactions with carrier, in comparison with diclofenac. Curcumin and diclofenac-loaded films showed great antibacterial activity against Gram-positive bacteria strains (*Bacillus subtilis* and *Staphylococcus aureus*, inhibition zone 16.67 and 13.67 mm, respectively), and in vitro and in vivo studies indicated that curcumin- and diclofenac-incorporated polymer films have great potential, as a new transdermal dressing, to heal wounds, because diclofenac can target the inflammatory phase and reduce pain, whereas curcumin can enhance and promote the wound healing process.

## 1. Introduction

As a specific biological process, wound healing refers to the growth and regeneration of tissues [1]. The wound healing process is considered to include five phases (hemostasis, inflammation, migration, proliferation, and maturation), where some of the phases may overlap [1,2,3]. The inflammatory phase is the first response to a skin injury, occurs immediately after the injury (together with the hemostatic phase) and lasts for approximately three days. During the inflammatory phase, various cellular and vascular processes stop further damage, eliminate pathogens and clean the wound [4]. After the inflammatory phase, the proliferation process takes place simultaneously with the migration process. During this phase, formation of granulation tissue, re-epithelization, and collagen synthesis by fibroblasts lead to wound damage repair [1,4].

A fluid called exudate is produced in the healing process and is present in almost all healing phases [5]. The produced exudate keeps the wound moist, which is an ideal environment for effective and efficient healing [6]. However, excess exudate can lead to complications in healing. Problems can also occur due to the appearance of pathogenic microorganisms (mainly bacteria and some strains of fungi) on the wound surface, which can cause severe infections and is often cited as the main reason for the prolonged healing process [1,7]. Furthermore, uncontrolled reproduction of pathogenic bacteria can lead to blood poisoning, sepsis, and even a fatal outcome [1,7]. From the above, it can be concluded that non-toxic and biocompatible formulations, with proper mechanical strength and capability to absorb excess exudate but prevent wound dehydration by maintaining a moist environment and effectively prevent or control infections, are among the key factors in wound treatment and healing and thus important tasks for the scientific community [8].

Bioactive or smart dressings are a class of dressings that are able to deliver bioactive compounds in the wound site and create an active and dynamic interaction with the wound’s environment [8]. Some formulations incorporating various bioactive molecules (dicarboxylic acids, Ag-nanoparticles, chitosan/propolis nanoparticles) that can prevent infection and have a positive effect on different wound healing phases and accelerate healing have been dealt with in numerous studies [1,9,10,11,12]. The use of dry and solid formulations that allow controlled release of the bioactive component over a longer period can give better therapeutic results because the patient is exposed to a drug concentration that is optimal for treatment [13]. Carriers can achieve incorporation and later the controlled release of various low-bioavailability drugs and antibiotics [13]. Controlled release of the bioactive component on the wound is achieved by carrier swelling during its contact with the wound exudate [13]. During swelling, the distance between the polymer chains increases, thus creating a system that can release drugs in a controlled manner. In addition to polymer hydration and swelling, a significant role in drug release from the carrier can be played by crosslinking the polymer within the carrier and the rates of potential carrier degradation and drug diffusion through the polymer matrix [13,14]. Recently, a hydrogel in the form of thin, elastic films has been increasingly applied in the wound healing process. This allows the unhindered transfer of matter from the film to the wound, secreting a moderate exudate amount [1]. Various natural (polysaccharides, proteins, and lipids) and synthetic polymers (poly(ethylene glycol), PEG, poly(vinyl alcohol), PVA, poly(ethylene oxide), PEO, and poly(vinyl pyrrolidone), PVP) can be used as constituents of these formulations [1,15]. Wound dressings can be developed from a combination of bio and synthetic polymers. Biopolymers suffer from poor mechanical properties that can be overcome by combining them with synthetic polymers. Carrageenan and alginate are constituents of wound dressing materials [16,17], but can form a stable formulation in combination with synthetic polymer poloxamer 407 able to incorporate hydrophobic bioactive compounds [18].

Diclofenac (Dlf) is a non-steroidal, anti-inflammatory drug (NSAIDs) and has the greatest application in the treatment of the painful rheumatic process [19]. This drug is commercially available in the form of various formulations for oral, dermal, or intramuscular application [19]. Oral administration of diclofenac is limited due to its low solubility in acidic media and possible diclofenac intramolecular cyclization [19]. To improve diclofenac bioavailability and to avoid its side effects after oral intake, diclofenac dermal application is more common, where possible [20]. Diclofenac does not affect individual phases (except inflammatory) in the wound healing process but indirectly affects healing because of its antibacterial properties [21,22]. Although the use of antibiotics to prevent infections is an effective solution, due to the occurrence of resistant pathogenic microorganisms and slower synthesis/isolation of new antibiotics, there is a need for alternative solutions [9]. Previous studies [23,24,25,26] have shown that diclofenac-incorporated formulations have antimicrobial activity, to some extent, against various bacterial strains. Since diclofenac primarily acts as an anti-inflammatory drug, which reduces post-injury pain, its additional advantage in wound treatment is its antibacterial property.

Curcumin (Cur) is a hydrophobic polyphenolic compound with antioxidant, anticancer, anti-inflammatory, and antimicrobial properties [27,28]. However, despite its high efficacy, the use of curcumin is limited due to its very low solubility and thus bioavailability [29]. For this reason, increasing curcumin bioavailability and developing formulations that serve that purpose have been the subjects of numerous studies [29,30,31,32]. Curcumin has good potential for wound healing treatment due to its antimicrobial, anti-inflammatory, and antioxidant properties. The wound healing process also includes reactive oxygen species, which are part of the immune response to the appearance of microorganisms [33]. However, prolonged exposure to reactive oxygen species in higher concentrations leads to oxidative stress, inhibiting the maturation phase during wound healing. For this reason, reactive oxygen species are the leading cause of prolonged inflammation [34,35,36]. Since curcumin has excellent antioxidant properties, great attention is paid to developing formulations that can be dermally applied, thus achieving the maximum anti-inflammatory effect of curcumin [37]. In addition, it has been found that curcumin can improve wound healing by participating in granulation tissue formation, damaged tissue regeneration, collagen deposition, thus improving epithelial cell regeneration processes and increasing fibroblasts proliferation [38]. Due to the above, more recent studies and review articles are dedicated to developing and describing various curcumin-containing wound healing dressings [39,40,41,42,43,44,45]. Polymer-based wound dressings (hydrogels, films, membranes, nanoparticles, nanofibers, liposomes) loaded with curcumin exhibited great in vitro and in vivo therapeutic outcomes [44]. Innovative strategies include formulations based on combination of curcumin with other anti-bacterial or anti-inflammatory agents [45,46].

In our previous research [18], the films based on carrageenan (Car), alginate (Alg) and poloxamer 407 (Pol) were optimized. These optimized films were used in this work to examine the efficiency of encapsulation and release of diclofenac individually and in a mixture with curcumin. After film characterization, the results obtained by diclofenac and curcumin release were related to the interactions these drugs achieve with the carrier components studied using theoretical models and AIM (Atoms in Molecules) analysis. Furthermore, considering the anti-inflammatory effect of curcumin [37], the synergistic effect of curcumin and diclofenac during the treatment of inflammation [47], as well as the positive effect of curcumin in the proliferation phase [38], films based on carrageenan, alginate and poloxamer containing a combination of these two drugs were prepared with the ultimate purpose of their application for in vivo wound healing.

## 2. Materials and Methods

### 2.1. Materials

Sodium alginate and κ-carrageenan were obtained from Roth (Karlsruhe, Germany). Curcumin, diclofenac (sodium salt), poloxamer 407, calcium chloride dihydrate, potassium chloride, sodium chloride, glutamine, fetal bovine serum, penicillin, streptomycin, resazurin, amoxicillin, tetracycline, 3-(4,5-dimethylthiazol-2-yl)-2,5-diphenyltetrazolium bromide (MTT), trypan blue, ketamine, and xylazine were purchased from Sigma-Aldrich (Burlington, MA, USA). Glycerol and ethanol were obtained from Honeywell (Charlotte, NC, USA). Sodium hydrogen phosphate dihydrate was purchased from Poch (Gliwice, Poland) and potassium dihydrogen phosphate from Kemika (Zagreb, Croatia). Non-essential amino acids were obtained from Capricorn Scientific GmbH (Ebsdorfergrund, Germany).

### 2.2. Film Preparation

Polysaccharides and poloxamer 407-based films (Car/Alg/Pol) were prepared by the casting method, using the procedure described in our previous research [18]. Appropriate polysaccharide masses (0.4 g of carrageenan and 0.1 g of alginate, total saccharide concentration 2.0% *w/w*) and the aqueous solution of poloxamer 407 (0.15 g, 5.0% *w/v*) were added to the aqueous solution of glycerol as plasticizer (60% *w/w* relative to total mass of saccharides) [18]. The mixture was stirred on a magnetic stirrer at room temperature for 1 h, then heated to 70 °C, and a solution of calcium chloride (0.5% *w/w*) was gradually added dropwise to the mixture (1 mL/min). After CaCl_2_ solution instillation, stirring was continued for 20 min under the same conditions, with further application of the ultrasonic bath. Then, the mixture was poured into Petri dishes (d = 9 cm) and dried for 20 h at a temperature of 40 °C. In the second phase, the dried semi-crosslinked films were immersed in a 10% glycerol and 3% calcium chloride solution for 10 min to achieve further crosslinking [18]. Finally, obtained crosslinked Car/Alg/Pol films were air-dried. To prepare films containing diclofenac (Car/Alg/Pol-Dlf), curcumin (Car/Alg/Pol-Cur), or a mixture of curcumin and diclofenac (Car/Alg/Pol-Cur+Dlf), an aqueous solution of diclofenac (1.0% *w/v*), i.e., a solution of curcumin in ethanol (1.0%, *w/v*), was added to the mixture of starting components (saccharides and poloxamer) after 30 min of initial stirring. Then, stirring was continued at room temperature for another 30 min, and the further work process was identical to that previously described.

### 2.3. Film Characterization

#### 2.3.1. Infrared Spectroscopy

FTIR (Fourier-transform infrared) spectra of films (Car/Alg/Pol, Car/Alg/Pol-Dlf and Car/Alg/Pol-Cur+Dlf) and starting components were recorded using infrared (IR) spectroscopy (Perkin Elmer Spectrum Two spectrophotometer, Waltham, MA, USA) to characterize the prepared film’s composition. The spectra were recorded in the range of 4000–500 cm^−1^.

#### 2.3.2. Texture Analysis

In order to investigate the mechanical properties of the prepared films, texture analysis was performed. The films were cut into a rectangular shape using a micrometer and a scalpel. The width of the samples was 10 mm and the gauge length was 30 mm, with a gripping length of 10 mm on each side. The thickness of each film was evaluated before tensile characterization, at five different points using the micrometer. The mechanical properties of the films were measured using a Texture Analyzer (TA.HD plus, Stable Micro Systems Ltd., Surrey, UK), equipped with a 5 kg load cell, using tensile grips A/TG. The test speed was 6 mm/s, with a trigger force of 0.09 N. The elongation at break (%EB) and tensile strength (TS) were estimated according to Equations (1) and (2), while Young’s modulus (YM) was estimated from the linear part of the stress–strain curve, according to the Equation (3). The time needed for the sample to break was also investigated.
(1)$$\mathrm{Elongation}\;\mathrm{at}\;\mathrm{break}=\frac{\mathrm{increase}\;\mathrm{in}\;\mathrm{length}\;\mathrm{at}\;\mathrm{break}}{\mathrm{initial}\;\mathrm{film}\;\mathrm{length}}\times 100$$
(2)$$\mathrm{Tensile}\;\mathrm{strength}=\frac{\mathrm{force}\;\mathrm{at}\;\mathrm{failure}}{\mathrm{cross}\;\sec\mathrm{tional}\;\mathrm{area}\;\mathrm{of}\;\mathrm{the}\;\mathrm{film}}$$
(3)Young's modulus=Δ StressΔ Strain 

The results of three replicates for each of the four films were expressed as the mean values ± SD.

#### 2.3.3. Scanning Electron Microscopy

Scanning electron microscopy (SEM) and energy dispersive X-ray (EDX) microanalysis were completed on the films using a Hitachi SU8030 instrument (Tokyo, Japan) with a field emission electron gun. The SEM is coupled with a Thermo Scientific NORAN System 7 detector (Madison, WI, USA) for X-ray microanalysis. Strips of each film were secured onto alumina stubs. Surface characterization was completed using a 1.0 keV accelerating voltage (V_a_) and a 10 µA emission current (I_e_) in the low magnification mode. Elemental point analysis was carried out at 20.0 keV (V_a_) and 10 µA (I_e_).

#### 2.3.4. XRD Analysis

X-ray diffraction (XRD) was used to evaluate the crystalline content of the films. Data were collected on a D8 Advance X-ray Diffractometer (Bruker, Germany) in theta–theta geometry in the transmission mode using Cu K_α_ radiation at 40 kV and 40 mA. A primary Göbel mirror for parallel beam X-rays and removal of Cu K_β_ radiation along with a primary 4° Soller slit, and a 0.2 mm exit slit was part of the setup. The sample rotation was set at 15 rpm, X-rays were collected using a LynxEye silicon strip position sensitive detector set with an opening of 3° with the LynxIris set at 6.5 mm and a secondary 2.5° Soller slit. Data collection was between 2 and 60° 2θ, step size of 0.02° and a counting time of 0.5 s per step. Two layers of the sample was secured between mylar film. Data were collected using DIFFRAC plus XRD Commander version 2.6.1 software (Bruker-AXS, Karlsruhe, Germany). Peak identification was completed using an EVA V6.0.0.7 (Bruker, Karlsruhe, Germany) software package.

#### 2.3.5. Thermogravimetric Analysis

Thermogravimetric analysis (TGA) was conducted using the Discovery 5500 TGA (TA Instruments, Crawley, UK) in aluminum pans with sample size 3.0 ± 0.5 mg for starting materials and 7.0 ± 1.0 mg for all film formulations. Samples were heated from ambient temperature (20 °C) to 500 °C at 10 °C/min, under nitrogen (25 mL/min). Data were analyzed using TA Advantage Universal Analysis V4.5 software (Lukens Dr, New Castle, DE, USA).

#### 2.3.6. Differential Scanning Calorimetry

Differential scanning calorimetry (DSC) was carried out using a Discovery 2500 DSC (TA Instruments, Crawley, UK) in hermetically sealed T zero aluminum pans with 3.0 ± 1.0 mg of sample. Sample was heated at 10 °C/min from −70 to 300 °C. Experiments were conducted in triplicate under nitrogen atmosphere (flow rate 50 mL/min). Data were analyzed using TA Advantage Universal Analysis V4.5 software (Lukens Dr, New Castle, DE, USA).

#### 2.3.7. Encapsulation Efficiency of Drugs

The encapsulation efficiency of curcumin and diclofenac was determined by immersing the films with incorporated drugs (Car/Alg/Pol-Dlf and Car/Alg/Pol-Cur+Dlf) in phosphate buffer pH 7.40. After 24 h, aliquots were taken, and the concentration of encapsulated drugs was determined using U*V/V*is spectrophotometry (Perkin Elmer U*V/V*is, Lambda 365, Waltham, MA, USA), at a wavelength of 430 nm for curcumin and 276 nm for diclofenac. The ratio of the spectrophotometrically determined drug weight to the weight of drug added to films in the preparation process represents the encapsulation efficiency (Equation (4)). The measurements were performed in triplicate.
(4)$$\mathrm{EE}\;\left(\%\right)=\frac{\mathrm{Spectrophotometrically}\;\mathrm{determined}\;\mathrm{amount}\;\mathrm{of}\;\mathrm{drug}}{\mathrm{Added}\;\mathrm{amount}\;\mathrm{of}\;\mathrm{drug}}\times 100$$

### 2.4. In Vitro Drug Release

The release of diclofenac from the Car/Alg/Pol-Dlf film as well as curcumin and diclofenac from the Car/Alg/Pol-Cur+Dlf film were monitored in vitro in conditions simulating wound exudate (PBS buffer, pH 7.4). For drug release testing, 2 × 2 cm films (diclofenac weight in the Car/Alg/Pol-Dlf film was 1.50 mg, while curcumin and diclofenac weight in the Car/Alg/Pol-Cur+Dlf film was 2.86 and 1.53 mg, respectively) were added to the buffer solution and incubated at 37 °C. Aliquots were taken at certain time intervals, and the concentrations of released diclofenac and curcumin were determined spectrophotometrically by measuring the absorbance at 276 and 430 nm, respectively. The measurements were performed in triplicate.

### 2.5. Drug Release Kinetics

Based on the results obtained during the in vitro release of drugs, the release kinetics were determined, indicating the mechanism of diclofenac and curcumin release from the films. The release kinetics were tested using various mathematical models, including zero-order kinetics, first-order kinetics, and the Higuchi, Hixon–Crowell, and Korsmeyer–Peppas release models, where $M_{t}/M_{\infty }$ represents the fraction of released drug at a given time (*t*) [48].
(5)$$\begin{array}{cc} \text{Zero-order kinetic:} & M_{t}/M_{\infty }=kt \end{array}$$
(6)$$\begin{array}{cc} \text{First-order kinetic:} & ln\;\left(M_{t}/M_{\infty }\right)=kt \end{array}$$
(7)$$\begin{array}{cc} \text{Higuchi model:} & M_{t}/M_{\infty }=kt^{1/2} \end{array}$$
(8)$$\begin{array}{cc} \text{Hixon–Crowell model:} & {\left(1-M_{t}/M_{\infty }\right)}^{1/3}=-kt \end{array}$$
(9)$$\begin{array}{cc} \text{Korsmeyer–Peppas model:} & M_{t}/M_{\infty }=kt^{n} \end{array}$$


A mathematical model that best describes the release of drugs from films can be determined based on the correlation coefficient (R^2^) value. Furthermore, the mechanism of drug release can be predicted based on the value of *n* (release exponent) [48].

### 2.6. Computational Details

Full geometry optimizations of the aggregate structures formed by attaching diclofenac and curcumin molecules to the drug carrier were performed at the semiempirical PM6 level of theory using the Gaussian 09 program package [49]. Structures of isolated curcumin and diclofenac molecules were optimized at the B3LYP/def2-SVP level of theory. All optimizations were done for six positions of two molecular systems (drug and carrier), which adopt face-to-face, side-to-side, and perpendicular arrangements, according to the scheme proposed in recent works [50]. Frequency calculations confirmed that the obtained optimized aggregate structures have no imaginary frequencies. Only the most stable structures were further examined.

In order to assess interactions between the drug molecule and its carrier, the binding energy (BE) was calculated through single point energy calculations at the B3LYP/def2-SVP level of theory. The BEs were computed as the difference between the B3LYP/def2-SVP electronic energy of the PM6 optimized aggregate structure and the sum of the B3LYP/def2-SVP electronic energies of the fragments whose geometries were extracted from the optimized aggregate structures. Van der Waals interactions in the studied complexes were estimated with Grimme’s D3 scheme [51] The AIM analysis was carried out by the Multiwfn program [52] and the obtained electron density of the bond critical points ($\rho (r_{BCP})$) was used to calculate the hydrogen bond binding energy (HBBE) as proposed by Emamian et al. [53]. In particular, the following equations were used:(10)$$HBBE=-223.08\times \rho (r_{BCP})+0.7423$$
(11)$$HBBE=-323.34\times \rho (r_{BCP})-1.0661$$
to calculate HBBEs for neutral and charged complexes, respectively.

### 2.7. Antibacterial Activity of Films

Antibacterial activity of the films (Car/Alg/Pol, Car/Alg/Pol-Cur, Car/Alg/Pol-Dlf, and Car/Alg/Pol-Cur+Dlf) was tested against four standard strains of bacteria. Antibiotic discs (A—amoxicillin 25 μg, T—tetracycline 30 μg, and S—streptomycin 10 μg) were used as positive controls. The experiment involved two Gram-positive bacteria (*Bacillus subtilis* ATCC 6633 and *Staphylococcus aureus* ATCC 25923) and two Gram-negative bacteria (*Pseudomonas aeruginosa* ATCC 27853 and *Escherichia coli* ATCC 25922).

*Bacterial suspensions—preparation and standardization*. Bacterial cultures were cultivated on nutrient agar before the experiment. The incubation period lasted 18–20 h at a temperature of 37 °C. The bacterial suspensions were prepared by the direct colony method. The procedure was performed under sterile conditions. First, 3–4 morphologically identical bacteria colonies were transferred to 5 mL of saline, mixed well to separate the cells and form a suspension. Then, the suspension turbidity was adjusted using a densitometer (DEN-1, BioSan, Latvia), McFarland 0.5 corresponding to 10^8^ CFU/mL. Bacterial suspensions were prepared immediately before the experiment, as they should be used approximately within 30 min of preparation [54,55].

*Disk diffusion method*. The susceptibility of bacteria to the tested films and standard antibiotics was tested by the in vitro disk diffusion method. The disk diffusion test was performed in a Petri dish on Mueller–Hinton (MH) agar (25 mL of medium per plate). Films and antibiotics discs were cut into cylinders measuring 5 mm in diameter. Films with tested substances and discs with specific concentrations of antibiotics were placed on the surface of the medium (3 identical films/discs on 1 plate), on which a pure bacterial suspension with 1–2 × 10^8^ CFU/mL was cultivated. After incubation (16–24 h), the inhibition zone diameter (the surface of the bacterial growth inhibition zone) was measured. The measured values were compared with the EUCAST standard [56], and the tested bacteria were classified as sensitive, moderately sensitive, and resistant [57]. All zones of inhibition were calculated in triplicates.

### 2.8. Cell Viability Study

*Cell culture*. In order to evaluate cell viability (proliferation) in the presence of the Car/Alg/Pol-Dlf and Car/Alg/Pol-Cur+Dlf films, a standard MTT test was applied [58]. A human fetal lung fibroblast cell line (MRC-5) was cultured in Dulbecco’s modified eagle medium supplemented with 10% fetal bovine serum, 100 U/mL penicillin, 100 μg/mL streptomycin, 2 mM L-glutamine, and 1 mmol/L non-essential amino acids. Cells were cultivated at 37 °C in an atmosphere of 5% CO_2_ and absolute humidity. The culture medium was completely replaced every 3 days, cell viability was determined using trypan blue staining, and only cell suspensions with viability greater than 95% were further used.

*Cell viability assay*. A viability study of the Car/Alg/Pol, Car/Alg/Pol-Dlf, Car/Alg/Pol-Cur+Dlf films was performed using an MTT assay. Firstly, the films were cut into cylinders of 9 mm in diameter. Secondly, the films were transferred into 96-well plates and irradiated by ultraviolet light for 30 min. Finally, the suspensions of MRC-5 cells (5000 cells per well, according to studies [40,59]) were dropped onto the sample surfaces. As a control, the same amounts of MRC-5 cells were dropped in the blank dishes. The plates were incubated for 24 and 48 h in an atmosphere of 5% CO_2_ and absolute humidity, at 37 °C. Then, MTT solution was added to cell culture and incubated. After incubation, MTT solution was removed, DMSO was added, and absorbance was measured at 595 nm with a multiplate reader. Experiments were performed in triplicates and repeated in three independent series.

### 2.9. In Vivo Study

All the animal research studies were approved by the Animal Ethics Committee of the Faculty of Medicine, University of Kragujevac (Ethical Approval Number: 01-6121). The use of the prepared films with incorporated mixture of curcumin and diclofenac and films containing only diclofenac was investigated for in vivo healing of burn-caused wounds. For in vivo study, male Wistar albino rats (6 to 8 weeks old, average body weight 200–250 g) were used. One group of animals (*n* = 3) was exposed to burns and not further treated (control), the second group (*n* = 3) was treated with the Car/Alg/Pol films, the third (*n* = 3) with the Car/Alg/Pol-Dlf films, and the fourth (*n* = 5) with the Car/Alg/Pol-Cur+Dlf films. The process of causing burns to rats was performed following the protocols in the previously published study [39]. Before causing burns, the animals were anesthetized with intraperitoneal ketamine (10 mg/kg body weight) and xylazine (5 mg/kg body weight). Then, the backs of healthy rats were shaved using depilatory cream. On the shaved skin area, the burns were caused by applying a hot metal plate (measuring 2 × 2 cm) to the skin for 10 s. Wounds caused this way were covered with the prepared films (measuring 2 × 2 cm). The healing process was monitored for seven days, with the daily replacement of film samples.

*Histopathological analysis*. The intensity of the skin injury caused by the hot metal plate was estimated based on histopathological analysis of healthy skin and skin exposed to burns. The contribution of incorporated drugs (diclofenac, curcumin) to the healing process was determined by comparing histopathological analyzes of untreated burned skin (control) and burned film-treated skin. All rats were sacrificed by means of cervical dislocation on day 7 post-burning. The skin was aseptically removed and fixed in 10% buffered formalin fixative overnight. Paraffin wax-embedded skin sections (5 µm) were stained with hematoxylin and eosin (H&E), and stained slides were then examined under a light microscope to evaluate the extent of damage. The images were captured with a light microscope equipped with a digital camera.

## 3. Results and Discussion

### 3.1. Film Characterization

#### 3.1.1. Basic Characteristics of Films

The average weights and thicknesses of the obtained films and the weight of drugs incorporated in films are shown in Table 1 (*n* = 5). It can be concluded from the obtained results that films of the same composition show uniformity in terms of both weight and thickness. According to the obtained results by statistical analysis, there were no significant differences between the mass and thickness of blank and drug-containing films (*p* > 0.05).

#### 3.1.2. FTIR Spectroscopy

The FTIR spectra of Car/Alg/Pol, Car/Alg/Pol-Dlf and Car/Alg/Pol-Cur+Dlf, as well as pure curcumin and diclofenac, are shown in Figure 1.

The spectra of pure drugs (both curcumin and diclofenac) show vibrational characteristic of aromatic C−C and C−H bonds can be observed. In addition, on the curcumin spectrum, the sharp band at 3508 cm^−1^ originates from the vibrations of the phenolic O−H group, and a band at 1628 cm^−1^ is the result of the valence vibrations of the C=O bond [60]. On the other hand, in addition to the vibrations of the bonds in the aromatic ring, the diclofenac spectrum is also characterized by bands at 1574 and 745 cm^−1^, which originate from the vibrations of the carboxylate anion and C−Cl bond, respectively [61].

The spectra of the Car/Alg/Pol, Car/Alg/Pol-Cur and Car/Alg/Pol-Cur+Dlf films, a wide absorption band in the range 3600–3000 cm^−1^ can be observed originating from the valence vibrations of the present saccharides −OH bonds. Additionally, valence vibrations of C−H bonds can be noticed in the range 3000–2840 cm^−1^. The width of the bands corresponding to the vibrations of the O−H bonds is a consequence of established hydrogen bonds. The FTIR spectrum of the Car/Alg/Pol film contains all group vibrations that are characteristic of both carrageenan (sulfate group vibrations—band at 1245 cm^−1^) and alginate (asymmetric and symmetric vibrations of carboxylate anion—bands at 1615 and 1417 cm^−1^), as well as poloxamer (C−O bond vibrations—very intense band at 1033 cm^−1^). From this, it can be concluded that a unique carrageenan/alginate/poloxamer hydrogel was formed. On the spectra of drug-containing films, bands characteristic of carrier constituents can also be observed. However, as a consequence of the addition and interactions of the drugs with the alginate from the carrier, there is a significant shift in the wavenumbers corresponding to the vibration of the carboxylate anion of diclofenac, 1574→1609 cm^−1^ (for the film Car/Alg/Pol-Dlf), or 1574→1602 cm^−1^ (for the film Car/Alg/Pol-Cur+Dlf). Due to the homogeneous drug distribution within the films, other characteristic bands of diclofenac and curcumin cannot be observed in the spectra of films containing these drugs.

#### 3.1.3. Texture Analysis

A desirable wound dressing should have good mechanical properties and maintain integrity during use [16]. A wound dressing should be flexible, elastic, and not prone to tear or rupture upon application, whether applied topically to protect dermal wounds or when used as an internal wound support [62]. The mechanical properties of the prepared films were evaluated using Texture Analyzer, and the results are presented in Table 2 as the mean values of three replicates for each film ± SD.

The mechanical strength of the prepared films was presented in terms of their tensile strength, the percentage of elongation to break (measure of extensibility), and Young’s modulus, a parameter used to describe the rigidity and stiffness of the material, as well. Comparison of the films Car/Alg/Pol and Car/Alg/Pol-Cur indicates that the addition of curcumin led to the slight decrease in elongation at break, and therefore decrease in extensibility. Time to break, tensile strength and Young’s Modulus values decreased, as well. Therefore, it can be concluded that addition of the curcumin to the films led to the slight decrease in material ductility and elasticity. The decrease in the strength and elongation of the break of the Car/Alg/Pol-Cur film can be a consequence of polymer–curcumin interactions on the films surface which results in crystals formation (can be also noticed by SEM analysis, Section 3.1.4). Similar results, in terms of %EB and TS reduction in the presence of curcumin, were obtained in the studies [63,64]. The results show that films containing diclofenac were stronger, stiffer, and less elastic than the Car/Alg/Pol-Cur films, as indicated by higher values of both TS and Young’s modulus and lower values of %EB. Compared to the blank films, similar conclusion can be obtained, with the note that the tensile strength is smaller for the Car/Alg/Pol-Dlf film. The addition of both curcumin and diclofenac was responsible for a small decrease in the elongation at break and time to break, whereas Young’s modulus increased in the comparison with blank films. Additionally, tensile strength was similar to blank films and higher in comparison with the Car/Alg/Pol-Cur and Car/Alg/Pol-Cur+Dlf films. The improvement in tensile strength of the transdermal Car/Alg/Pol-Cur+Dlf films might be attributed to the high aspect ratio and rigidity which results from the strong affinity between the polymers and drugs. For all drug containing films, the decrease in the extensibility could be attributed to the restriction of mobility of polymer chains in the presence of drugs due to strength of polymer–drug interactions (explained in the Section 3.4).

#### 3.1.4. SEM Analysis

Secondary electron images (magnification ×500) captured from the four prepared films show clear difference in the surface morphology (Figure 2).

The Car/Alg/Pol film (Figure 2a) had a smooth, homogeneous, and uniform surface, indicating the excellent film formability of carrageenan, alginate and poloxamer. Meanwhile, the Car/Alg/Pol-Cur film presented somewhat uneven surfaces (Figure 2b). Some convex pieces and crystalline particles were observed on the surface when the curcumin was added to the film. Observation of the surface of Car/Alg/Pol-Cur (Figure 2b) reveals crystals that are embedded into the polymer matrix. This can be explained by the fact that hydroxyl groups presented in carrageenan, alginate and curcumin could dehydrate and condense with the carboxyl or sulfate group from polysaccharides. The complex reaction can result in the formation of a complex three-dimensional network structure. Similar results were obtained in the study of Xie et al. [65], where curcumin-loaded chitosan/pectin films were developed. The Car/Alg/Pol-Dlf film (Figure 2c) has a uniform and smooth surface, but contrast difference is visible because of the presence of diclofenac, in comparison with the Car/Alg/Pol film surface. Additionally, the surface of diclofenac-incorporated film also had some white spots and fine dust-like particles, which had been reported that the calcium ions could accumulate and form white patches in the polysaccharide film [65]. The morphological characteristics of the surface of the Car/Alg/Pol-Cur+Dlf film (Figure 2d) had more similarities with the Car/Alg/Pol-Cur film surface than with Car/Alg/Pol-Dlf, due to higher concentration of incorporated curcumin in comparison with diclofenac. The presence of crystalline particles and small dust-like particles covering the surface can be noticed at Figure 2d. Lower concentration of diclofenac and its higher solubility enabled more evenly distribution of diclofenac in the film network, in comparison to curcumin.

EDX analysis was carried out to identify the elemental composition of polymer matrix (Figure 2a), as well as drug-loaded films (Figure 2b–d). It was proved that the elemental structure of Car/Alg/Pol was mainly comprised of carbon (25.3 w%) and oxygen (61.2 w%), while the remaining mass is made up of sodium, sulfur, chlorine, calcium and potassium. The results are in concordance with chemical structure of polymers presented in the carrier (carbon and oxygen, as the main constituents of carbohydrates and poloxamer; sodium, potassium and chlorine, as usual impurities and counterions of alginate and carrageenan; sulfur, as a constituent of carrageenan and calcium as crosslinking ion). The EDX results of drug-loaded films showed similar results with the ones for the blank films. Addition of diclofenac could be confirmed by chlorine content increase, in comparison with the Car/Alg/Pol and Car/Alg/Pol-Cur films, whereas curcumin addition has no significant influence on elemental analysis due to its chemical composition.

#### 3.1.5. XRD Analysis

The physical form of the films was determined using X-ray diffraction. The obtained diffractograms for the prepared films are shown in Figure 3.

Curcumin and diclofenac, as pure substances, are presented exclusively in crystalline form [18,23]. All the prepared films are predominantly amorphous, based on obtained XRD patterns, with the note that the Car/Alg/Pol-Cur and Car/Alg/Pol-Cur+Dlf films also have crystalline content. The diffractogram corresponding to the Car/Alg/Pol-Dlf film indicates the exclusively amorphous state of all present components, which means that diclofenac molecules have dispersed within the carrier. These results can be related to the results obtained by SEM analysis (Section 3.1.4), where crystalline particles are present on the surface of curcumin-containing films, while the Car/Alg/Pol-Dlf film exhibited smooth surface. In the case of the Car/Alg/Pol-Cur and Car/Alg/Pol-Cur+Dlf films, the presence of two crystalline peaks can be observed, which can be attributed to one of the curcumin polymorphs [66]. This can be explained by the conversion of one structure of curcumin to the other polymorph form (which differ from each other in the keto-enol orientation of curcumin molecules) during film preparation or its storage [66].

#### 3.1.6. Thermogravimetric Analysis

The thermal stability of the films was studied using thermogravimetric analysis. Figure 4 shows the thermograms corresponding to the Car/Alg/Pol, Car/Alg/Pol-Dlf, Car/Alg/Pol-Cur and Car/Alg/Pol-Cur+Dlf films, as well as pure curcumin and diclofenac.

Based on the obtained results, it can be noticed that thermograms corresponding to the films are similar, in terms of the weight loss stages. On the other hand, pure curcumin and diclofenac have one significant mass loss at 250 and 280 °C, respectively, related to their thermal decomposition. The initial mass loss in all films (around 10%), which occurs at temperatures below 100 °C, is caused by the evaporation of weakly bound water (remained after drying during film preparation) or water absorbed from the air, present in the sample. Therefore, the films have a tendency to absorb water.

The next significant mass losses occur in the range of temperatures from 150 to 280 °C, in two stages. The first weight loss at a temperature above 150 °C is the result of evaporation of glycerol, presented in films as a plasticizer [67]. This process is followed by alginate decomposition that starts at approximately 220 °C and continues up to 240 °C [68]. As a consequence of these processes, films lost approximately 26% of their mass. The second significant weight loss (calculated mass loss is approximately 15%) is related to the carrageenan decomposition that starts at approximately 250 °C and takes place up to 280 °C [69]. Finally, the last mass loss (approximately 22%) occurred due to poloxamer thermal decomposition in the temperature range of 380 to 400 °C [70]. Due to small concentrations of curcumin and diclofenac in the films, their decomposition is not clearly noticeable in thermograms; as it coincides with decomposition of carrageenan and alginate.

#### 3.1.7. Differential Scanning Calorimetry

With the aim to investigate the drug physical state and polymers behavior in the films, DSC analysis of prepared formulations and starting materials was carried out at a temperature from −70 to 300 °C, as shown in Figure 5.

The DSC thermograms show a complex thermal behavior of starting materials and the film formulations. Carrageenan and alginate DSC thermograms did not show any thermal events at temperatures below 200 °C or sharp endothermic peaks above it, which confirms their predominantly amorphous nature. The presence of complex endothermic peak at 208 and 202 °C for carrageenan and alginate, respectively, can be linked to the thermal decomposition of the polymers. On the other hand, on DSC thermogram of poloxamer, an endothermic (melting) peak is present at 54 °C in addition to an exothermic peak at 163 °C, which could be attributed to poloxamer recrystallization from the melt [23,70]. Diclofenac, at higher temperatures, showed a significant exothermic peak, which was immediately followed by two endothermic peaks, which are result of diclofenac melting and its thermal decomposition [71,72]. The DSC thermogram of curcumin revealed a single sharp peak at 179 °C, which corresponds to the melting point of crystalline curcumin [70].

At higher temperatures, resemblance to the degradations processes in carrageenan and alginate is evident in all the prepared films, which can be seen in Figure 5b. The Car/Alg/Pol film showed an altered endothermic peak at 230 °C, suggesting that the thermal characteristics of polymers changed during film production, which is caused by polymer interaction, similar to the results obtained by Boateng et al. [23]. In the DSC thermogram which corresponds to the Car/Alg/Pol-Dlf film, only broad peaks can be observed, thus confirming its amorphous structure. Due to the interaction of polymers and diclofenac, which led to diclofenac transformation from crystalline to amorphous state, peaks attributed to its melting and decomposition cannot be noticed. On the other hand, endothermic peak which corresponds to biopolymers thermal decomposition can be observed at 220 and 225 °C, for the Car/Alg/Pol-Cur and Car/Alg/Pol-Cur+Dlf films, respectively. Additionally, curcumin-containing films showed two small endothermic peaks at 167 °C, which can be attributed to the melting point of curcumin polymorphs [66], similar to the results obtained by XRD analysis (Section 3.1.5).

#### 3.1.8. Drug Encapsulation Efficiency

In the previous research propylene glycol nanoliposomes containing curcumin were developed for burn wound healing with the encapsulation efficiency of 84.66% [73]. Additionally, natural (chitosan), synthetic (poly-lactic co-glycolic acid) and semi-synthetic (carboxymethylcellulose) polymer-based nanoparticles were used for curcumin delivery with high encapsulation efficiencies (higher than 90%) [74]. Previously published articles also studied polymer-based (chitosan and alginate/carboxymethyl chitosan/aminated chitosan) carriers with high diclofenac encapsulation efficiency (84 and 95%, respectively) [75,76]. The percentage of drug encapsulation in our study was defined by determining the weight of drugs (diclofenac and the mixture of curcumin and diclofenac) incorporated into the films. Polysaccharide- and poloxamer-based carriers easily interact with the added drugs, forming a unique, homogeneous film. The encapsulation efficiency of diclofenac in the Car/Alg/Pol-Dlf film is (92.65 ± 3.20)%, while the encapsulation efficiency of curcumin and diclofenac in the Car/Alg/Pol-Cur+Dlf film is (90.49 ± 3.90)% for Cur and (98.83 ± 4.25)% for Dlf. Results revealed that the tested drugs are incorporated within the films in a high percentage, which further leads to an increase in their bioavailability.

### 3.2. In Vitro Release Study

The in vitro study of diclofenac release from the films Car/Alg/Pol-Dlf (Figure 6) demonstrate that a high percentage of release was achieved at the beginning (initial burst in the first 15 min), and continues to grow gradually in a period of up to 3 h. Subsequently, it begins to stabilize within 24 h, with a final release percentage of (90.10 ± 4.89)%. By comparing the results obtained in this study with the results of other studies [23,24], where it was also monitored the release of diclofenac from polymer-based films, it can be concluded that a higher release percentage in 24 h is achieved in our work compared to the results of other works, where the release percentage after 72 h was 60% [23,24].

By studying the profiles of drug release from films containing a mixture of curcumin and diclofenac (Figure 7), it can be noticed that a slightly higher percentage of diclofenac release was obtained compared to the release results obtained from films of the same composition containing only diclofenac (Figure 6). Additionally, the final release percentage of diclofenac (95.61 ± 1.67)% after 24 h is greater than the curcumin release rate (90.48 ± 0.30)% over 24 h. The graphs that follow the drugs release process (Figure 7) are similar to the previously obtained graph corresponding to the release of individual drugs: diclofenac (Figure 6), or the graph obtained in the study monitoring the release of curcumin from films of the same composition [18]. Therefore, it can be concluded that curcumin and diclofenac generally retain their individual characteristics during the release process when mixed within the carrier, Car/Alg/Pol-Cur+Dlf.

Comparing the results obtained in our paper with the percentage of diclofenac release in the presence of *Curcuma longa* plant extract from the transdermal gel [77] reveals the advantages of prepared polysaccharide- and poloxamer-based films. The percentage of diclofenac release after 24 h was 84.19% [77], which is lower compared to the percentage achieved in our study (95.61%). Additionally, a study by Mendes et al. [78] investigated phospholipid nanofibers, based on polysaccharide chitosan, for transdermal delivery of individual drugs diclofenac and curcumin. The percentages of curcumin and diclofenac release after 24 h were approximately 20% and 60%, respectively [78], while after 7 days, the maximum release of 75% was achieved for curcumin and 80% for diclofenac [78]. Therefore, it can be concluded that our investigation gave significantly better results in terms of the efficiency of the prepared carriers.

Figure 7 shows that drug release is cumulative over 24 h. It can also be seen that curcumin is released from the carrier at a much slower rate in the initial hours compared to diclofenac, which indicates stronger curcumin and carrier interactions. In addition to drug interactions with the carrier, swelling of the carrier itself, diffusion of the solute, and carrier degradation are crucial factors influencing the release of drugs from polymeric carriers [79]. Since the solubility of curcumin in buffer is significantly lower than the solubility of diclofenac, its diffusion rate into buffer solution will also be lower. Therefore, the solubility of curcumin directly causes its slower release compared to the release of diclofenac.

In vitro release results are in accordance with the fact that the local reduction in the inflammation response is advocated in burn management. Fast diclofenac release is preferable because anti-inflammatory drugs suppress a persistent inflammatory response, leading to improved wound healing [80,81]. Biphasic pattern of diclofenac release involving the two stages is targeted to control the local inflammation and pain associated with a burn injury. Thus, the burst anti-inflammatory drug release effect ensures both a rapid reduction in painful sensation and the management of the pro-inflammatory mediators’ cascade released at the burn level and is needed immediately after lesion occurrence [82]. After mentioned burst release, the gradual drug delivery phase offers an anti-inflammatory and analgesic local effect over the longer period needed for burn healing. Diclofenac release profile obtained in this work (especially from the Car/Alg/Pol-Cur+Dlf film) is desirable for burn treatment as the first 12 h are critical and correspond to the peak of the inflammatory phase [83]. On the other hand, release of curcumin, as a drug which accelerate different phases of wound healing, should be in a sustained manner, similar to the release profiles obtained in previous studies [46,73]. Release mechanism which includes a burst release of antibacterial drug diclofenac followed by a sustained release of curcumin with stronger antibacterial activity is expected to be effective in controlling and preventing infection in the very early stages of wound infliction. Prolonged curcumin release might indicate a long scale antimicrobial potency fabricated biocomposite dressings [84].

### 3.3. Drug Release Kinetics

The equations corresponding to zero-order kinetics, first-order kinetics, as well as the Higuchi, Hixon–Crowell and Korsmeyer–Peppas release models were applied to the results obtained by in vitro diclofenac and curcumin release study to investigate the mechanism of drug release from films. The correlation coefficient values obtained by fitting the results in accordance with the equations corresponding to the models are shown in Table 3. Additionally, the values of the release rate constants are shown and the value of *n* in the Korsmeyer–Peppas model.

The release of curcumin, as well as diclofenac from the Car/Alg/Pol-Dlf and Car/Alg/Pol-Cur+Dlf films, is best described with the Hixon–Crowell model, which is otherwise characteristic of systems in which the release rate is largely controlled by drug solubility in buffer rather than by diffusion of particles through the matrix [48]. Still, it should be considered that curcumin solubility is significantly lower than that of diclofenac, which further explains the slower curcumin release from the carrier. Curcumin release can also be described with the Highuchi model, which is characteristic of drugs with particles dispersed within a uniform, solid matrix, which acts as a diffusion medium, where drug release is largely controlled by Fick’s law of diffusion [48]. These mechanisms of drug release from the tested formulations were further confirmed using the Korsmeyer–Peppas model, which describes very well the release of curcumin and diclofenac from all tested films (high correlation coefficient values were obtained).

The Korsmeyer–Peppas model is significant for estimating the mechanism of drug release and is mainly used to determine the parameter that has the greatest impact on the release rate (polymer swelling, diffusion of incorporated substance, polymer erosion) [48]. The value obtained for the release exponent *n* of 0.5 directly indicates the release controlled by drug diffusion, while the value of *n* = 1 indicates that drug release occurs primarily due to polymer swelling [48]. If the values of *n* differ from the above, then the release mechanism is influenced by several factors. In general, values of the release exponent below 0.5 correspond to Fick’s law-controlled diffusion, above 0.5 to diffusion that does not obey this law, where release is also caused by polymers erosion, and values above 1 correspond to the super-transport case [48].

Considering the results obtained using the Korsmeyer–Peppas model, and following the values of *n*, it can be concluded that the diclofenac mechanism of release is the same during its release from the film containing only diclofenac (*n* = 0.14) and from the film containing a mixture of curcumin and diclofenac (*n* = 0.15). The obtained low value of the release exponent indicates that diclofenac release is primarily controlled by its diffusion from the carrier into the buffer, explaining the high release rate. Additionally, this result is in agreement with the Hixon–Crowell model, which describes the release of diclofenac from these formulations. The value of the curcumin release exponent from the film containing a mixture of curcumin and diclofenac is 0.68, which indicates that the release mechanism is influenced by the diffusion rate and swelling of the polymer. The obtained results are in agreement with the description of curcumin release using Higuchi and Hixon–Crowell models.

### 3.4. Theoretical Study of Component Interaction in Developed Films

In this section, the experimentally obtained results were further rationalized by means of quantum chemical computations. The optimized, most stable structures of the complexes formed between the diclofenac and curcumin molecules and the corresponding drug carriers are displayed in Figure 8 and Figure 9. The most stable structures of the complexes were obtained by curcumin binding to carrageenan, as well as diclofenac binding to alginate from drug carrier. It should be noted that the curcumin-based structure presented in Figure 9 is somewhat different from that found in our previous study [18]. In the present work a more stable aggregate structure was obtained in which the curcumin molecule better adopts the shape of the drug carrier, thus maximizing bonding interactions.

It was found that the bonding interactions between carboxylate anion of diclofenac and hydroxyl groups of alginate from carrier are dominated by two hydrogen bonds, of which the hydrogen bond 1 is found to be very strong. Based on Equation 11 the binding energies of hydrogen bonds 1 and 2 in the diclofenac containing complex are found to be −38.9 and −9.5 kcal/mol, respectively. These results are also in concordance with results obtained by FTIR analysis (Section 3.1.2) and wavenumber shift of carboxylate anion in films containing diclofenac, as a consequence of formation of hydrogen bonds between diclofenac carboxylate anion and alginate. On the other hand, in the case of curcumin there are two rather weak hydrogen bonds (formed between phenolic group/oxygen from ether group of the drug and carrageenan from carrier). According to Equation 10 the HBBE for bonds 1 and 2 in the curcumin-based complex are −7.3 and −1.7 kcal/mol, respectively.

The BEs calculated at the B3LYP/def2-SVP level of theory are given in Table 4. The obtained BEs predict that the bonding interactions between diclofenac and the drug carrier are more pronounced than curcumin-carrier interactions. From the optimized aggregate structures shown in Figure 8, it can be anticipated that bonding interactions between diclofenac and the drug carrier are manly determined by the strength of the formed hydrogen bonds, whereas curcumin binds through much more pronounced dipol-dipol and van der Waals interactions (Figure 9). Relevance of the van der Waals interactions in the case of curcumin comes from the fact that this molecule, in comparison with diclofenac, has much wider molecular surface and much more flexible geometry which enables an efficient adsorption on the drug carrier surface.

The BEs calculated with a more appropriate theoretical treatment, which accounts dispersion interactions (characteristic for curcumin) through Grimme’s D3 method, show that curcumin is much stronger bonded to the carrier than diclofenac (Table 4). It should be pointed out that these results are in agreement with the experimentally obtained in vitro release data, which shows that diclofenac can be easier released from carrageenan/alginate/poloxamer carrier than curcumin.

### 3.5. Antibacterial Activity of Films

All tested bacteria showed sensitivity to the tested antibiotic disks (amoxicillin, tetracycline, streptomycin) prescribed in the manufacturer’s instructions and the valid EUCAST standard [56]. Gram-negative bacteria: *Pseudomonas aeruginosa* ATCC 27853 and *Escherichia coli* ATCC 25922 did not show sensitivity to any of the tested films. The resistance of Gram-negative bacteria to the effect of films containing components with antimicrobial potential can be explained by the carrier’s structure. Carrageenan is an anionic polysaccharide, and since it is present in a large percentage in the carrier, it can be assumed that the carrier also will carry a negative charge. In addition, Gram-negative bacteria contain an additional outer layer composed of negatively charged lipopolysaccharides. Therefore, it is believed that more positively charged carriers will have a greater antimicrobial effect against these bacteria strains because they can achieve more favorable electrostatic interactions. On the other hand, films prepared in our work are expected to have a more pronounced tendency against Gram-positive bacteria.

The Car/Alg/Pol carrier and the Car/Alg/Pol-Cur film did not show antibacterial activity against the tested bacterial strains. In contrast, the Car/Alg/Pol-Dlf and Car/Alg/Pol-Cur+Dlf films were active against the Gram-positive bacteria strains. Table 5 shows the zones of bacterial inhibition obtained by the effect of the prepared Car/Alg/Pol-Dlf and Car/Alg/Pol-Cur+Dlf films, as well as antibiotics (amoxicillin, tetracycline and streptomycin) as controls, on strains of *Bacillus subtilis* ATCC 6633 and *Staphylococcus aureus* ATCC 25923.

As can be seen from the presented results, films containing a mixture of curcumin and diclofenac give significantly better results compared to films containing only diclofenac as the active substance. In the case of the Car/Alg/Pol-Cur+Dlf film, the antibacterial activity of both diclofenac and curcumin is pronounced. The obtained results can be related to the results achieved by monitoring the in vitro release of the mixture of drugs. Films containing only curcumin did not exhibit antimicrobial activity against the tested bacteria, despite curcumin’s favorable antibacterial properties. This can be explained by the fact that the present bacteria reproduce much faster compared to the rate of antibacterial agent curcumin release from the carrier. Experimental results indicate that diclofenac is rapidly released from formulations, approximately 50% in the first 30 min, while only 2% of curcumin is released during this time [18]. Even though diclofenac shows low antibacterial activity (compared to commercially available and used antibiotics – amoxicillin, tetracycline, and streptomycin), its concentration in the initial phase of release is high enough to slow the growth of bacteria. Thanks to the fast action of diclofenac, conditions for the further antibacterial action of curcumin were created despite its slow release. As can be seen from the attached results (Table 5), the inhibition zone of the Car/Alg/Pol-Cur+Dlf film for bacteria *Bacillus subtilis* ATCC 6633 corresponds to the inhibition zone induced by the antibiotic amoxicillin and is similar to the inhibition zone provided by the antibiotic streptomycin. For strain *Staphylococcus aureus* ATCC 25923, the zone of inhibition of the applied film is only slightly smaller than the zone of inhibition caused by the antibiotic streptomycin.

The antimicrobial activity of the film with a mixture of drugs against Gram-positive bacteria can also be related to the results of the study by Adamczak et al. [85] that investigated the antimicrobial activity of curcumin on more than 100 strains of pathogens. The results indicated that the susceptibility of Gram-positive bacteria was significantly higher than that of Gram-negative bacteria. The susceptibility of the species was not related to its genus. It was concluded that curcumin has great potential as a very selective antibacterial agent [85]. Other studies [86,87,88,89] confirmed the synergistic antibacterial activity of curcumin with different antibiotics (cefaclor, cefodizime, cefotaxime, gentamicin, amikacin, and ciprofloxacin) against different strains of bacteria. The antimicrobial effect of diclofenac in the presence of antibiotics was also examined [23]. Films with incorporated streptomycin (30%, *v/v*) and diclofenac (10%, *v/v*) were prepared to be used for faster healing of chronic wounds. The application of films enabled the controlled release of streptomycin and diclofenac for 72 h. Films with incorporated drugs gave higher inhibition zones against *Staphylococcus aureus*, *Pseudomonas aeruginosa*, and *Escherichia coli* compared to zones of inhibition provided by pure drugs. Still, the concentration of drugs used in the films was very high [23].

Our work studied the synergistic effect that occurs in the combination of curcumin with the non-steroidal anti-inflammatory drug diclofenac. Based on all the above, it can be concluded that the film Car/Alg/Pol-Cur+Dlf has great potential for treating infections caused by strains of *Bacillus subtilis* ATCC 6633 and *Staphylococcus aureus* ATCC 25923 as its action is similar to the effect of antibiotics (amoxicillin, streptomycin). Additionally, the side effects of the Car/Alg/Pol-Cur+Dlf film (resistance and undesired effects in the gastrointestinal tract) are expected to be significantly lower compared to that of commercially available antibiotics.

### 3.6. Cell Viability Assay and In Vivo Wound Healing Study

The effect of Car/Alg/Pol, Car/Alg/Pol-Dlf, and Car/Alg/Pol-Cur+Dlf on MRC-5 cell line viability was examined by MTT test after cultivation for 24 and 48 h (Figure 10).

The obtained results indicate that the film Car/Alg/Pol has no effect on the healing process as the percentage of examined cells viability is the same as in the control sample. Additionally, after 24 h of incubation, in the presence of films containing diclofenac, it is observed that the viability of MRC-5 cells is higher compared to the control, but without a statistically significant difference. However, cell viability is significantly higher in the presence of films containing a mixture of curcumin and diclofenac during the incubation period of 24 and 48 h. Comparing the obtained results with the results obtained in the study of curcumin films [18], it can be concluded that, after 24 h, the increase in cell viability occurs exclusively due to the presence of curcumin in films since similar viability percentages were obtained for films containing only curcumin and a mixture of diclofenac and curcumin. Based on the percentage of viable cells, it can also be concluded that a film containing a mixture of curcumin and diclofenac may have a potential for application in the wound healing process. The presence of diclofenac, although it does not contribute to the healing process, does not interfere with the positive effect of curcumin. On the other hand, the film containing a mixture of drugs shows significant antibacterial activity. This additionally demonstrates its suitability for use since, in addition to affecting the inflammation and proliferation phases in the wound healing process, it also prevents infections. The obtained positive results of in vitro assay have directed further research, and the prepared films were tested as formulations potentially applicable for wound healing of the skin of rats.

*Histopathological analysis*. A representative photomicrograph of rats’ skin sections from all groups (control, treated, and non-treated) stained with H&E is shown in Figure 11.

Analysis of paraffin sections of skin tissue of untreated animals, stained by hematoxylin-eosin technique, showed a healthy skin structure. Epidermis, dermis, and subcutaneous adipose tissue had normal histological structure, with intact sweat and sebaceous glands. In addition, the structure of hair follicles is preserved. A sample of burned skin tissue showed clear signs of epidermis and dermis damage compared to the untreated group of animals. The analysis showed damage to all epidermis layers—infiltration of inflammatory cells is observed in the dermis, indicating inflammatory skin changes. Additionally, heavy bleeding in the dermis was recorded. In the group treated with the Car/Alg/Pol films, disorganization of all epidermis layers and infiltration of inflammatory cells with minimal signs of re-epithelialization were observed. Histopathological analysis of burned skin tissue showed minimal tissue regeneration signs in animals treated with the Car/Alg/Pol-Dlf films, including reduced infiltration of inflammatory cells compared to the group treated with only the Car/Alg/Pol films. In contrast to the above films, a notable degree of skin regeneration was observed in the group treated with the Car/Alg/Pol-Cur+Dlf films. On the sections of burned skin tissue treated the with Car/Alg/Pol-Cur+Dlf films, significant regeneration of the epidermis is observed, with well-organized layers and minimal infiltration of inflammatory cells. Results of histopathological analysis can be related to the cell viability study because the obtained results indicated that the Car/Alg/Pol-Cur+Dlf film enhances cell proliferation. Considering both in vitro and in vivo data, our findings clearly demonstrate that the application of films, containing both curcumin and diclofenac, improves the healing of burns remarkably. Available data from the literature related to the dermatological effects of curcumin, mentioned above, indicate its anti-inflammatory and antioxidant effect achieved through enhanced synthesis of hyaluronic acid and the effect of increasing skin moisture [35,36,37,38,44], along with the known anti-inflammatory effect of diclofenac [19]. The presented drug characteristics and the results obtained through our in vivo study led to a conclusion that the use of curcumin and diclofenac films achieves effective healing of burn-caused dermal wounds.

## 4. Conclusions

In biocompatible films based on polymers κ-carrageenan, alginate and poloxamer, diclofenac (an anti-inflammatory drug that has antibacterial properties) as well as a mixture of curcumin (a drug that exhibits antioxidant, anti-inflammatory and antibacterial properties) and diclofenac were incorporated. The characterization of the films showed that the prepared films have a smooth, homogeneous surface, while XRD analysis indicated a decrease in the crystallinity degree of curcumin and diclofenac after their incorporation into the films, while diclofenac transforms to an amorphous state. The in vitro release study showed that the bioavailability of curcumin and diclofenac was significantly improved by using developed carriers. Based on the results obtained by drug release kinetics, it was concluded that the polymer swelling degree has the greatest influence on curcumin release, while the release of diclofenac is largely controlled by diffusion. Theoretical examination of interactions that carriers establish with curcumin and diclofenac indicated that diclofenac formed strong hydrogen bonds with alginate from the carrier, while curcumin established stronger, primarily dispersion interactions with carrageenan. The antibacterial study of the prepared films showed that films with diclofenac and a mixture of curcumin and diclofenac inhibit the growth and development of Gram-positive bacteria *Bacillus subtilis* and *Staphylococcus aureus*. Additionally, it was determined that drug- loaded films are not cytotoxic, whereas films containing a mixture of curcumin and diclofenac can increase cell viability and thus have a favorable effect on cell proliferation, which is a phase during wound healing. Based on the results of in vivo study, it can be concluded that the produced films have great potential for healing wounds caused by burns.

## Figures and Tables

**Figure 1 polymers-14-04091-f001:**
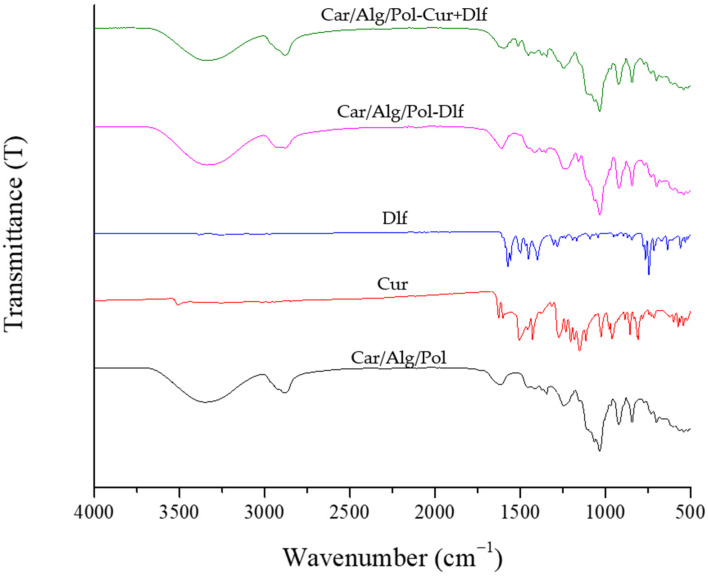
FTIR spectra of curcumin, diclofenac, and the Car/Alg/Pol, Car/Alg/Pol Dlf and Car/Alg/Pol-Cur+Dlf films.

**Figure 2 polymers-14-04091-f002:**
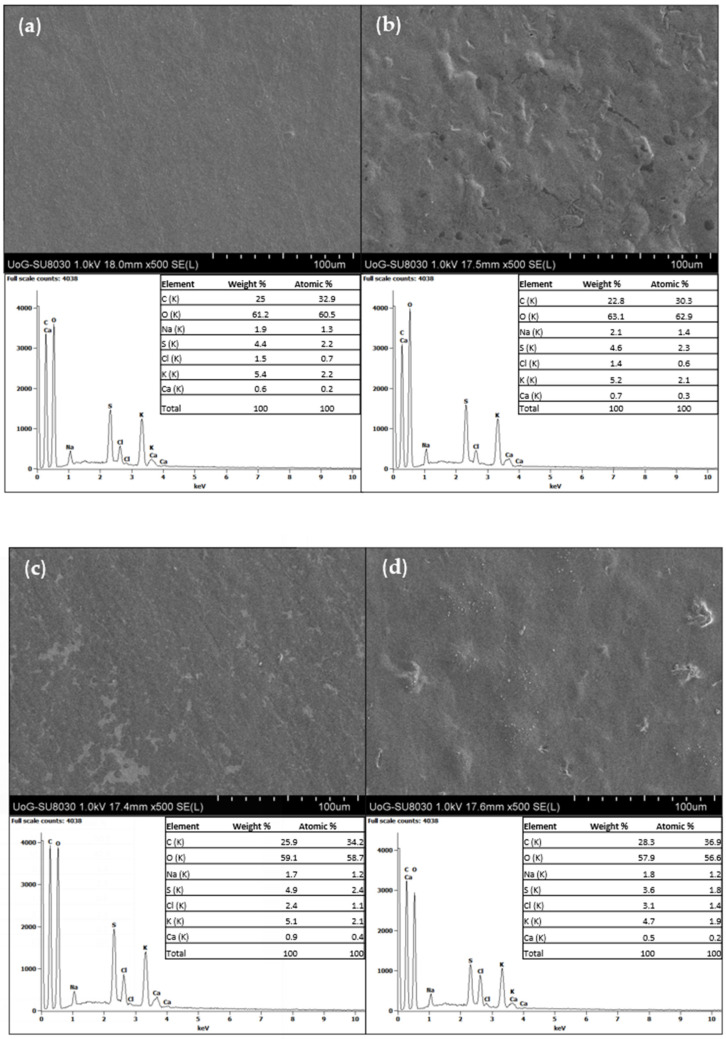
SEM images (×500) and EDX analysis of the (**a**) Car/Alg/Pol, (**b**) Car/Alg/Pol-Cur, (**c**) Car/Alg/Pol-Dlf, (**d**) Car/Alg/Pol-Cur+Dlf films.

**Figure 3 polymers-14-04091-f003:**
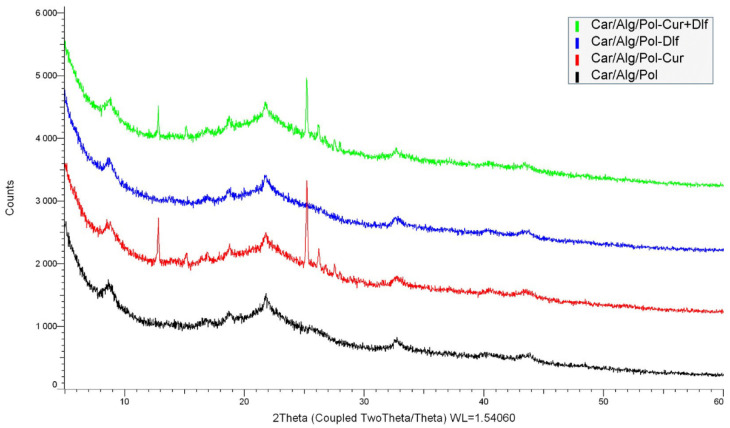
XRD patterns of the prepared films.

**Figure 4 polymers-14-04091-f004:**
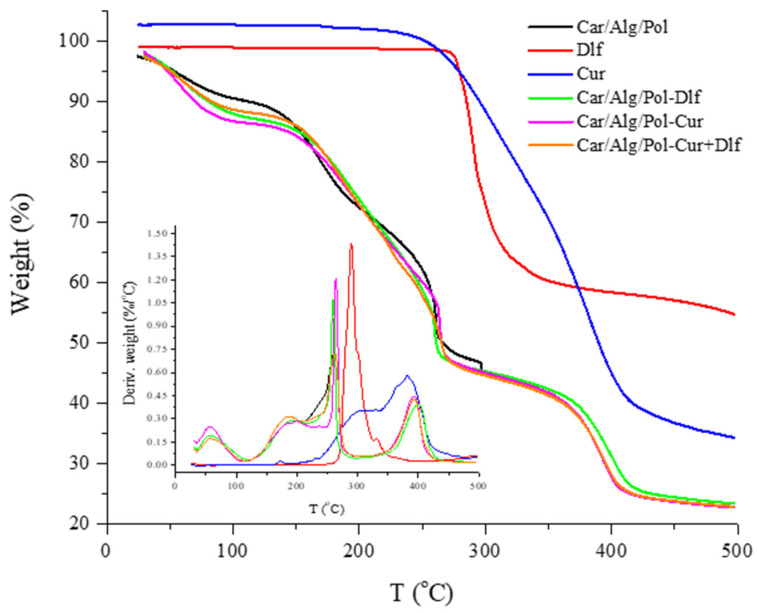
TGA and DTG curves of diclofenac, curcumin, and the Car/Alg/Pol, Car/Alg/Pol-Cur, Car/Alg/Pol-Dlf and Car/Alg/Pol-Cur+Dlf films.

**Figure 5 polymers-14-04091-f005:**
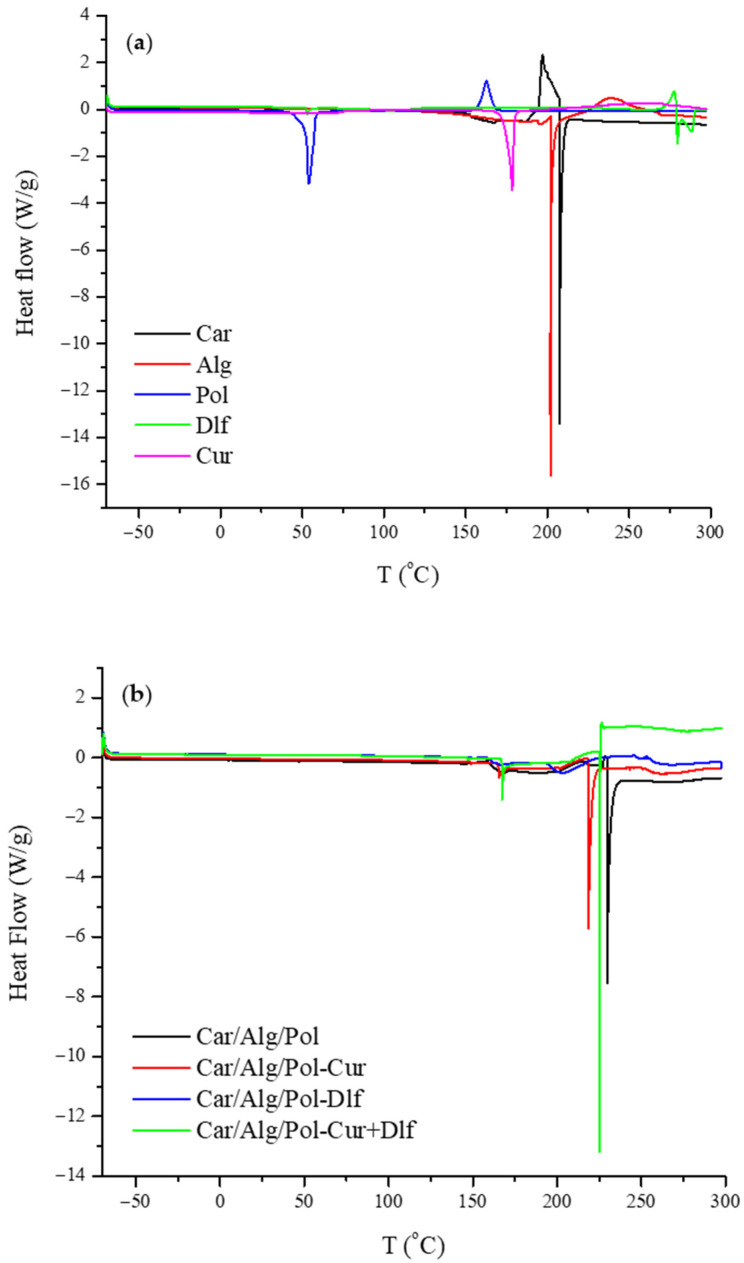
DSC thermograms of (**a**) starting components and (**b**) prepared films.

**Figure 6 polymers-14-04091-f006:**
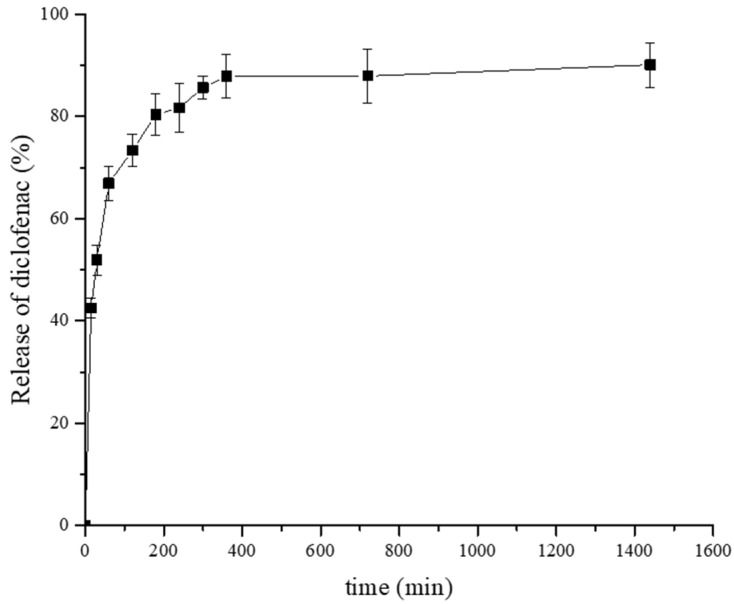
In vitro release of diclofenac from the Car/Alg/Pol-Dlf film (*n* = 3).

**Figure 7 polymers-14-04091-f007:**
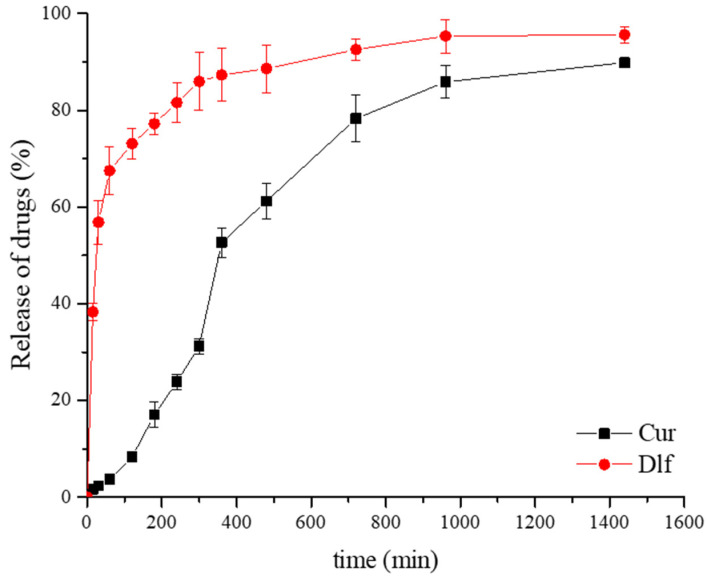
Curcumin and diclofenac in vitro release from the Car/Alg/Pol-Cur+Dlf film (*n* = 3).

**Figure 8 polymers-14-04091-f008:**
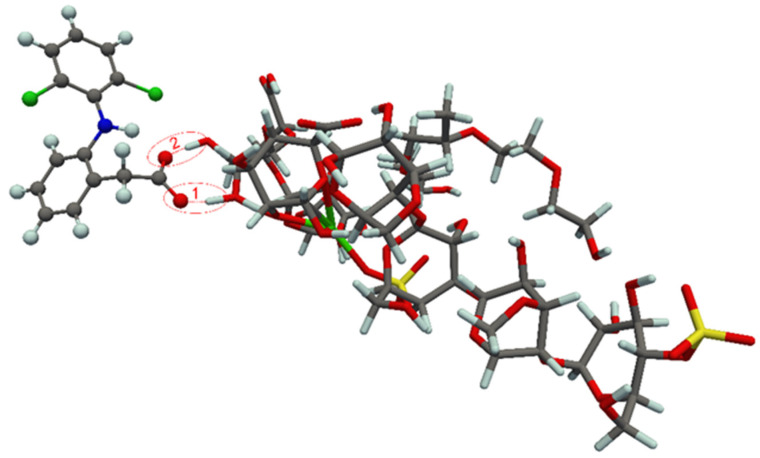
Optimized structure of diclofenac-carrier complex. Hydrogen bonds are denoted by red dashed lines (1 and 2). For the sake of clarity, the ball-and-stick and licorice visualization models were used for molecular structures of the drug and drug carrier, respectively.

**Figure 9 polymers-14-04091-f009:**
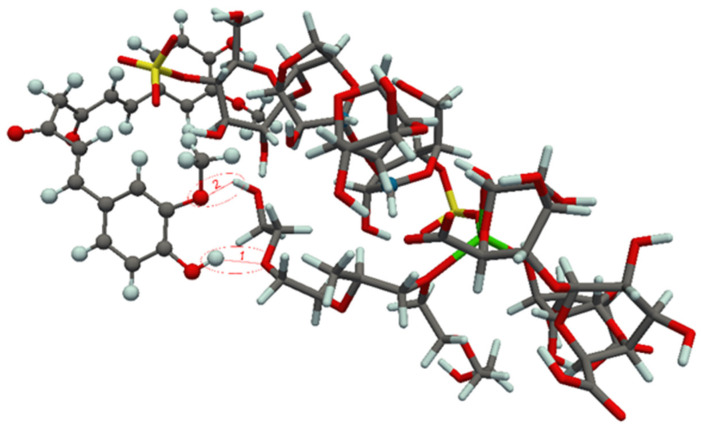
Optimized structure of curcumin-carrier complex. Hydrogen bonds are denoted by red dashed lines (1 and 2). For the sake of clarity, the ball-and-stick and licorice visualization models were used for molecular structures of the drug and drug carrier, respectively.

**Figure 10 polymers-14-04091-f010:**
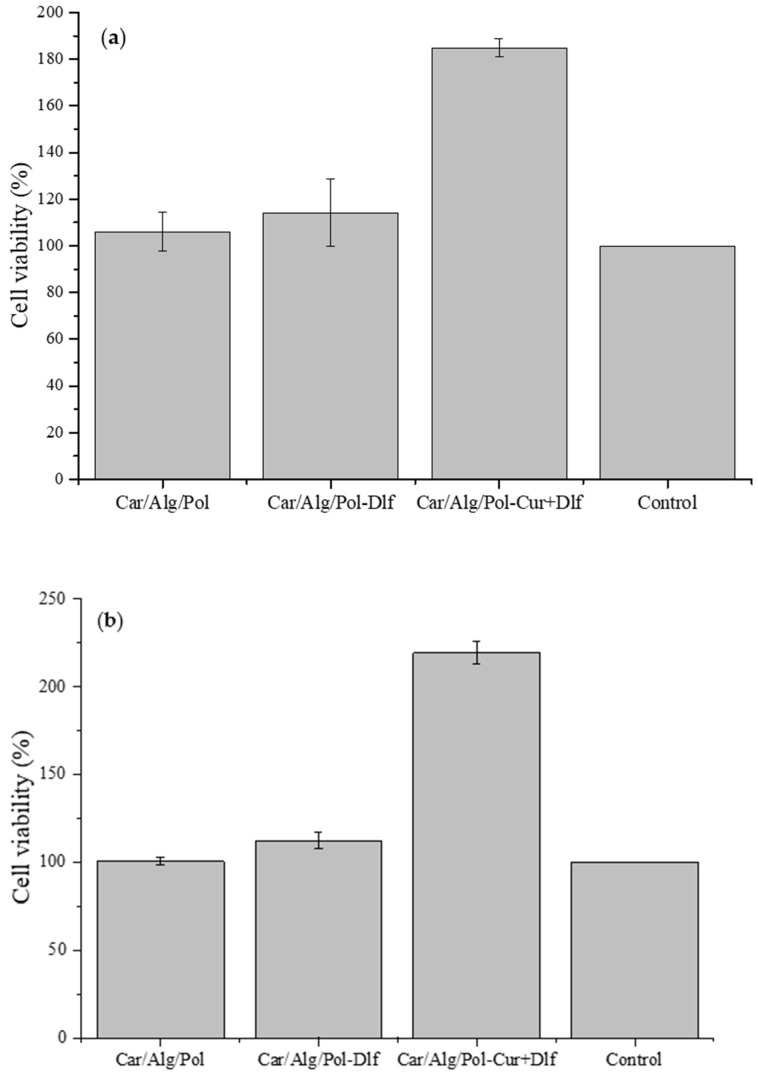
Influence of the Car/Alg/Pol, Car/Alg/Pol-Dlf i Car/Alg/Pol-Cur+Dlf films on MRC-5 cell viability after (**a**) 24 h and (**b**) 48 h.

**Figure 11 polymers-14-04091-f011:**
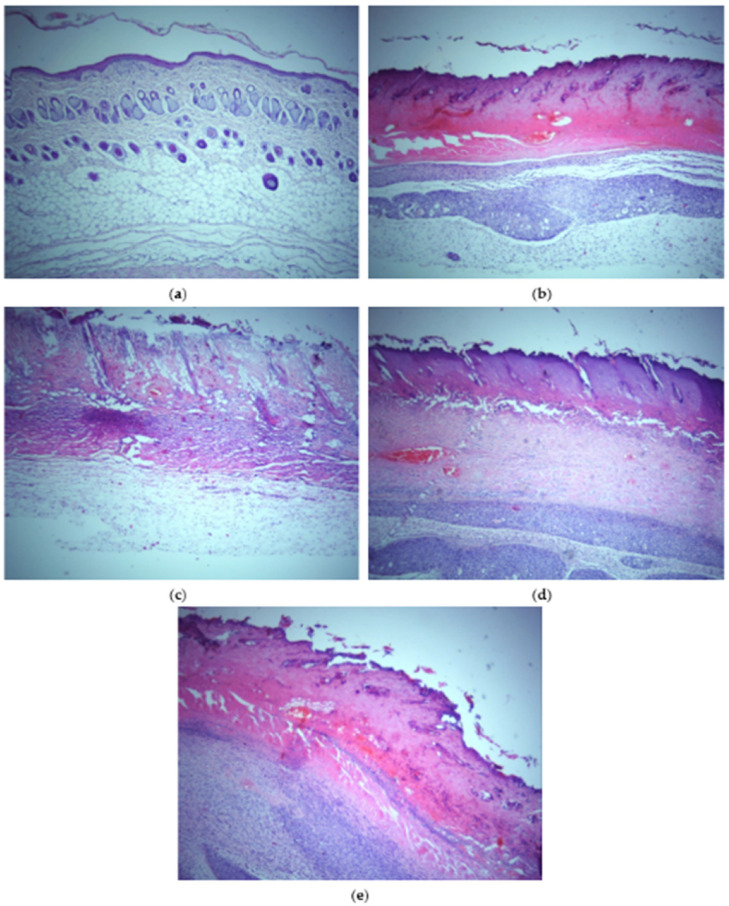
Histopathological observation of H&E stained skin sections (200×) of (**a**) healthy skin, (**b**) burned skin, (**c**) burned skin treated with the Car/Alg/Pol film, (**d**) burned skin treated with the Car/Alg/Pol-Dlf film, and (**e**) burned skin treated with the Car/Alg/Pol-Cur+Dlf film.

**Table 1 polymers-14-04091-t001:** Basic film characteristics (*n* = 5).

Film	Mass of Film (mg/cm^2^)	Film Thickness (μm)	Mass of Drug (mg/cm^2^ of Carrier)
Car/Alg/Pol	12.21 ± 0.65 [18]	104.27 ± 3.35 [18]	/
Car/Alg/Pol-Dlf	13.79 ± 0.65	121.12 ± 0.93	0.375 ± 0.012
Car/Alg/Pol-Cur+Dlf	14.29 ± 0.13	134.83 ± 2.17	0.718 ± 0.028 (Cur)
			0.400 ± 0.017 (Dlf)

**Table 2 polymers-14-04091-t002:** The mechanical properties of analyzed films (*n* = 3).

Sample Name	Elongation at Break (% ± SD)	Tensile Strength (MPa ± SD)	Young’s Modulus (MPa ± SD)	Time to Break (s ± SD)
Car/Alg/Pol	32.41 ± 1.02	34.60 ± 1.31	4.00 ± 0.04	3.09 ± 0.13
Car/Alg/Pol-Cur	27.36 ± 4.20	27.62 ± 2.63	3.86 ± 0.20	2.69 ± 0.40
Car/Alg/Pol-Dlf	25.19 ± 3.40	28.14 ± 1.63	4.41 ± 0.43	2.66 ± 0.39
Car/Alg/Pol-Cur+Dlf	29.66 ± 3.38	32.66 ± 0.18	4.87 ± 0.43	2.77 ± 0.34

**Table 3 polymers-14-04091-t003:** Values of correlation coefficients, release rate constants and release exponent.

Film	Zero-Order Kinetics		First-Order Kinetics		Highuchi Model	
	*k_0_*	R^2^	*k_I_*	R^2^	*k_H_*	R^2^
Car/Alg/Pol-Dlf	0.0156	0.3896	0.0202	0.3357	0.1083	0.6372
Car/Alg/Pol-Cur+Dlf (Cur)	0.0483	0.8529	0.1567	0.6112	0.2716	0.9370
Car/Alg/Pol-Cur+Dlf (Dlf)	0.0182	0.5347	0.0241	0.4304	0.1159	0.7541
	**Hixon** **−Crowell Model**		**Korsmeyer** **−Peppas Model**			
	*k_HC_*	R^2^	*k_KP_*	*n*		R^2^
Car/Alg/Pol-Dlf	0.0328	0.7955	0.7412	0.1400		0.8466
Car/Alg/Pol-Cur+Dlf (Cur)	0.0431	0.9930	0.1427	0.6807		0.9213
Car/Alg/Pol-Cur+Dlf (Dlf)	0.0321	0.9283	0.6861	0.1529		0.9069

**Table 4 polymers-14-04091-t004:** Binding energies (BEs in kcal/mol) for diclofenac and curcumin in the respective complexes calculated the B3LYP/def2-SVP and B3LYP-D3/def2-SVP levels of theory.

Method	Diclofenac	Curcumin
B3LYP/def2-SVP	−36.5	−28.5
B3LYP-D3/def2-SVP	−44.9	−56.9

**Table 5 polymers-14-04091-t005:** Results of susceptibility of tested Gram-positive bacteria to films and antibiotics discs.

Bacteria	*Bacillus subtilis* ATCC 6633	*Staphylococcus aureus* ATCC 25923
**Tested films**	**Inhibition zone (mm) ∑**	
Car/Alg/Pol-Dlf	9.33	6.33
Car/Alg/Pol-Cur+Dlf	16.67	13.67
**Antibiotic disks**	**Inhibition zone (mm) ∑ and sensitivity category (S ^1^)**	
A	16.67—S	26.67—S
T	29.67—S	23.33—S
S	18.33—S	16.67—S

^1^ S—sensitive.

## Data Availability

The data are contained within this article.

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
