# Peer review of "Curcumin and Diclofenac Therapeutic Efficacy Enhancement Applying Transdermal Hydrogel Polymer Films, Based on Carrageenan, Alginate and Poloxamer"

_polymers, 2022, doi:10.3390/polym14194091_

Round 1

Reviewer 1 Report

Manuscript entitled “Curcumin and diclofenac therapeutic efficacy enhancement applying transdermal hydrogel polymer films, based on carrageenan, alginate and poloxamer” could be interesting for the readers. However, the paper needs a major revision before publication. I have listed a few comments that need to be addressed:

1.       Add some more concrete results in the abstract. Remove the name of characterization methods.

2.       Introduction could be much better with more background about the work with up-to-date citations. Also, I would advise to cite few recent reviews article on this topic and curcumin added films in the introduction part.

3.       What is the novelty of this review work?

4.       Why Car/Alg/Pol was chosen? Add reason.

5.       Add a schematic diagram to show the overall work.

6.       Write the full form once when mentioning for the first instance.

7.       SEM image is not clear, improve the quality.

8.       Add DTG in Fig. 4

9.       Add statistical analysis significant difference in Table 1.

10.    Why cell viability increased from 100 to 300 % that is very strange? Need clear explanation.

11.    The integration of the results from different parameters should be improved carefully.

12.    The obtained results were not discussed well with previously published literature. So, the discussion part needs much improvement, and more up to date citation should be added.

13.    Conclusion is long, make it short and to the point.

14.    Also, carefully revise the typos and linguistic error to make the manuscript error free.

Reviewer 2 Report

The assessed manuscript “Curcumin and diclofenac therapeutic efficacy enhancement applying transdermal hydrogel polymer films, based on carrageenan, alginate and poloxamer” present a comprehensive investigation on anti-inflammatory and antibacterial agents; diclofenac and curcumin encapsulated within composite hydrogel films for wound healing applications.

Being under the scope of this journal, this research can contribute potential significant knowledge to the field.

There are some points below to be addressed:

1.      The methods section is need to be clarified in terms of reproducibility. Therefore, it is recommended to add exact quantities of reagents used in the experiments.

2.      When you mention the details of supplied materials or used devices: Please mention the city and country information of the manufacturer/supplier in the parenthesis.

3.      In Figure 1, axis titles are not clear. It is recommended to be written full rather than showing only symbol/units, such as ‘Transmittance (T, %)’ and ‘Wavelength (cm-1)’.

4.      In the line -799, there is a supportive information from the literature. The citation to this reference is missing so it should be added.  This sentence can be revised for clarity to prevent confusion between the results obtained in the current study and the information in the literature.

5.      It could be beneficial to discuss what is the suitable release profile and total release time of antibacterial and anti-inflammatory drugs to ensure an efficient wound healing process. This is recommended to be discussed under Results and discussion section.
